# An Intelligent Method for Lead User Identification in Customer Collaborative Product Innovation

**Jiafu Su** [1,2,3], **Xu Chen** [2], **Fengting Zhang** [1], **Na Zhang** [4] **and Fei Li** [2,5,*]

1   Research Center for Economy of Upper Reaches of the Yangtze River, Chongqing Technology and Business University, Chongqing 400067, China; jeff.su@cqu.edu.cn (J.S.); 2020658005@email.ctbu.edu.cn (F.Z.)
2   School of Management and Economics, University of Electronic Science and Technology of China, Chengdu 610054, China; chenx@uestc.edu.cn
3   International College, Krirk University, Bangkok 10220, Thailand
4   School of Mines, China University of Mining and Technology, Xuzhou 221116, China; zhangna@cumt.edu.cn
5   CISDI Engineering Co., Ltd., Chongqing 400013, China
*   Correspondence: qinling1987@hippoedu.net

**Abstract:** For customer collaborative product innovation (CCPI), lead users are powerful enablers of product innovation. Identifying lead users is vital to successfully carrying out CCPI. In this paper, in order to overcome the shortcomings of traditional evaluation methods, a novel intelligent method is proposed to identify lead users efficiently based on the cost-sensitive learning and support vector machine theory. To this end, the characteristics of lead users in CCPI are first analyzed and concluded in-depth. On its basis, considering the sample misidentification cost and identification accuracy rate, an improved cost-sensitive learning support vector machine (ICS-SVM) method for lead user identification in CCPI is further proposed. A real case is provided to illustrate the effectiveness and advantages of the ICS-SVM method on lead user identification in CCPI. The case results show that the ICS-SVM method can effectively identify lead users in CCPI. This work contributes to user innovation literature by proposing a new way of identifying highly valuable lead users and offers a decision support for the efficient user management in CCPI.

**Keywords:** lead user; customer collaborative product innovation; lead user identification; support vector machine; cost-sensitive learning

## 1. Introduction

Innovative ideas generated by users are of great value, because their ideas have greater novelty and user value compared to the ideas of employees [1–3]. Since Von Hippel [4,5] put forward the concept of "lead users", the important group of lead users has been valued by academia and business. Lead users possess rich professional product knowledge and product experience, and they can accurately catch the development trend of market demand and provide valuable product innovation ideas [1,6]. Firms such as IBM, Apple, Microsoft, Cisco, Huawei, and Xiaomi increasingly invest in the customer collaborative product innovation (CCPI) mode to solicit users' ideas and contributions during their product innovation processes. The trend of CCPI represents a shift from a closed, internal R&D process toward an open collaborative innovation that can provide more novel and diverse insights into users' needs and problem-solving abilities [7–9]. In the CCPI process, different users have different knowledge, abilities and intentions, which can generate different product innovation contributions and values [10]. Among them, the lead users have advanced demands and highly expected profitability. On the one hand, they have advanced needs that will become widespread in the target market, while other general users may only have these needs after several months or years. On the other hand, they can obtain great material or spiritual benefits from product innovation solutions that meet their advanced needs [11,12]. The emergence of the first chromatograph, surgical navigation

system, and neurosurgical medical robots are all examples of successful CCPI [13,14]. In sum, CCPI has strong business potential, which can improve the diversity of products and help enterprises to achieve breakthrough innovation.

CCPI does not mean that all users should participate in the product innovation process; it is mainly initiated by the lead users. Therefore, if a firm wants to carry out CCPI, it must identify lead users in order to integrate their knowledge and ideas into product innovation process. Through the identification of lead users, firms can enable them to participate in product innovation, design, trials, and other work directly related to user needs [15,16]. In this way, it can greatly improve the speed of product development, ensure that products meet user demand, and reduce product innovation risk. Additionally, due to the limited product innovation resources of firms, firms cannot provide all users with equal innovation resources to carry out CCPI activities [17]. Therefore, it is of great practical significance for firms to identify lead users as their important product innovation partners.

Lead user identification is the process of recognizing and locating the users with the attributes of lead users from a large user group, and then making them join in the CCPI process. The accurate identification of lead users is the premise and basis of effective user management. Currently, two major efforts of lead user identification have been addressed to improve identification effectiveness and efficiency: the lead user characteristics [18] and the identification methods [1,19]. As for the lead user characteristics, Von Hippel developed the lead user concept drawing on observations of the user innovation phenomenon [4,5] and concluded the typical characteristics of lead users. However, he did not propose a quantitative method to search for or identify lead users. Bettencourt et al. [20] stated that the helping behavior and personal initiative are important for lead user identification. Fuller et al. [21] pointed out that user motivation is of greater value in identifying lead users. Ye [22] stated that collaboration willingness, product knowledge, and brand recognition are important characteristics of lead users and established a comprehensive evaluation model of lead user identification. Tang et al. [23] constructed the lead user identification index system and established the fuzzy comprehensive evaluation model of lead user identification.

The main methods of lead user identification are divided into two categories: traditional identification methods, such as scanning method [24], pyramid method [25], fuzzy comprehensive evaluation method [26]; and intelligent identification methods of machine learning based on wavelet neural networks [27] and support vector machines [28]. Berm and Bilgram [1] proposed the netnography and crowdsourcing method to identify lead users in social media. Kratzer et al. [29] investigated the positions of lead users in social networks and stated that the social network perspective can provide a helpful reference to other lead user identification methods. Fu et al. [30] stated that positive emotional and self-interest-oriented linguistic styles are significantly positively related to lead-userness, and then proposed a text mining method to identify lead users. The above works provide some helpful ideas for the identification of lead users in CCPI. Moreover, we can also find that traditional evaluation methods such as fuzzy comprehensive evaluation involve the calculation of index weights with subjectivity, which cannot ensure the robustness of identification results of lead users. In addition, traditional evaluation methods rely on questionnaires or interviews to obtain data, which are time-consuming and labor-intensive. Especially when the number of candidate users is huge, they are not only difficult to implement, but also make it difficult to quickly and accurately identify lead users.

However, intelligent identification methods can not only make full use of the existing historical data of firms, but also avoid the problem of strong subjectivity in the process of identification. Therefore, at present, more and more firms are beginning to use intelligent machine learning methods to identify lead users, that is, to learn and train the user sample data that firms have already acquired in the process of CCPI. Then, when inputting the corresponding data of new user identification factors, the lead users who meet the corresponding characteristic factors in other user groups or new user groups can be identified quickly [23,30].

Among intelligent machine learning methods, the support vector machine (SVM) method is widely used for its excellent binary classification performance [31]. The standard support vector machine method has good classification performance on class-balanced sample sets, while its classification performance on class-unbalanced sample sets is relatively poor. Moreover, SVM is generally based on the principle of structural and empirical risk minimization, and it has the same default misclassification costs [32,33]. However, if a firm misidentifies an ordinary user as a lead user, it pays a certain amount of management costs, but it is still able to quickly identify that said user is not a lead user in the process of CCPI. Conversely, if a firm misclassifies a lead user as an ordinary user, it loses a lot of opportunity cost due to the great value of lead users to CCPI. Therefore, the opportunity cost that a firm loses when a few samples of lead users are misidentified is higher than the cost when ordinary users are misidentified. To fill this research gap, the cost-sensitive learning (CS) theory is introduced into the SVM method to solve the problems of unbalanced sample size and misclassification cost inconsistency in this work [34,35]. Moreover, an improved cost sensitive learning and support vector machine (ICS-SVM) method is proposed to decrease the misidentification cost and error rate of lead user identification, which can provide valuable decision support to identify lead users in CCPI.

## 2. The Factor Sets for Lead User Identification

In the model of lead user identification, the input is the factor data of users, and the output is the identification results of the different types of users. Synthesizing the existing research of lead user characteristics [5,26], lead user identification should not only consider common attributes such as a user's gender and age, but also consider behavior attributes, knowledge attributes, and ability attributes in CCPI. Based on a comprehensive investigation of existing research, this paper has sorted, processed, and refined the characteristics of the lead user identification in CCPI. Moreover, this work proposes that the factors of lead user identification should include three types of factors: user general attributes, user activity attributes, and user knowledge level attributes, and the corresponding sub-factors are also further proposed. The factors of lead user identification and their quantitative methods are described in detail as follows.

### 2.1. The Factors of User General Attributes

The factors of users' general attributes reflect user personal general information, which include user preference and support for the brand product from one side. Specifically, the factors of users' general attributes conclude gender (X1), age (X2), monthly income (X3), number of owned products from a brand (X4), and investment in purchasing products from a brand (X5). The gender factor can be quantified by Boolean variables, where male is 1 and female is 0. The quantification of other factors can use their corresponding specific values. In order to unify the dimension, the data of other factors need to be normalized before constructing the basic data, except for the data of gender factors. The normalization method is shown below:

$$x_i = \frac{x_i - x_i^{\min}}{x_i^{\max} - x_i^{\min}} \tag{1}$$

where $x_i^{\max}$ is the maximum value of all data in factor $x_i$, and $x_i^{\min}$ is the minimum value of all data in $x_i$.

### 2.2. The Factors of User Activity Attributes

User activity is the behavioral attribute of a user when they participate in the CCPI process. Due to the development of information technology, the activity space of users has shifted more to the internet. Furthermore, many brand online user communities have formed. In brand online user communities, users can easily exchange experiences or help each other with certain products. The activity of users in brand online user communities is a very important indicator to identify lead users [15,30].

The activity of users' participation in online communities can draw on the RFM model in the marketing field of analyzing user response and user value. Specifically, the RFM model includes recency (R), frequency (F), and monetary value (M) [36,37]. For a user, "recency" represents the length of time since the user's last purchase of a certain brand product, "frequency" denotes the total number of purchases within a specified time period, and "monetary value" means the amount of money spent on the certain brand products in this specified time period. For the identification of lead users in CCPI, the RFM model is used to measure user activity. Moreover, due to the differences in specific application areas, the meanings of the three indicators of the traditional RFM model first need to be revised and quantified.

Recency (X6)

The recency variable indicates how long it has been since the user last posted or followed a post in the online community; its time unit is day. For user $i$, its recency can be obtained as:

$$R_i = Tn - T_i \tag{2}$$

where $Tn$ is the current date and $T_i$ is the last time of participation in the online community.

Frequency (X7)

The frequency variable indicates how often a user participates in online community activities. For user $i$, its frequency can be obtained as:

$$F_i = N/T_i' \tag{3}$$

where $N$ is the total number of times the user has participated in posting and following posts in the online community, and $T_i'$ is the length of time (in days) over which the user has joined in the online community activities.

Monetary (X8)

The monetary value variable indicates the impact of a user on other users in the online community. In social network theory, centrality is an indicator of the actual authority and influence of individuals in the entire network, which includes degree centrality, closeness centrality, and betweenness centrality [38]. Among them, degree centrality is the most effective indicator that can be used to reflect the monetary value. Degree centrality is closely related to the influence of users in an online community [39]. Users with a higher degree centrality will also have a greater impact on other users. In social network theory, degree centrality refers to the number of other nodes connected to one node. In the CCPI online community, the monetary value of a user is the number of other users who are connected to the user, that is, the user's number of friends in the online community.

In order to unify the dimension, the related data of user activity also need to be normalized before constructing the basic data. The normalization method is the same as Equation (1).

### 2.3. The Factors of User Knowledge Attributes

User knowledge is another important factor in identifying lead users. In CCPI, lead users should have a certain level of knowledge. By combining the research of lead user knowledge and the needs of CCPI knowledge systems, this paper extracts the factors of knowledge for lead user identification, including basic knowledge factors, demand-based knowledge factors, and innovative skills factors. The details are shown in Figure 1.

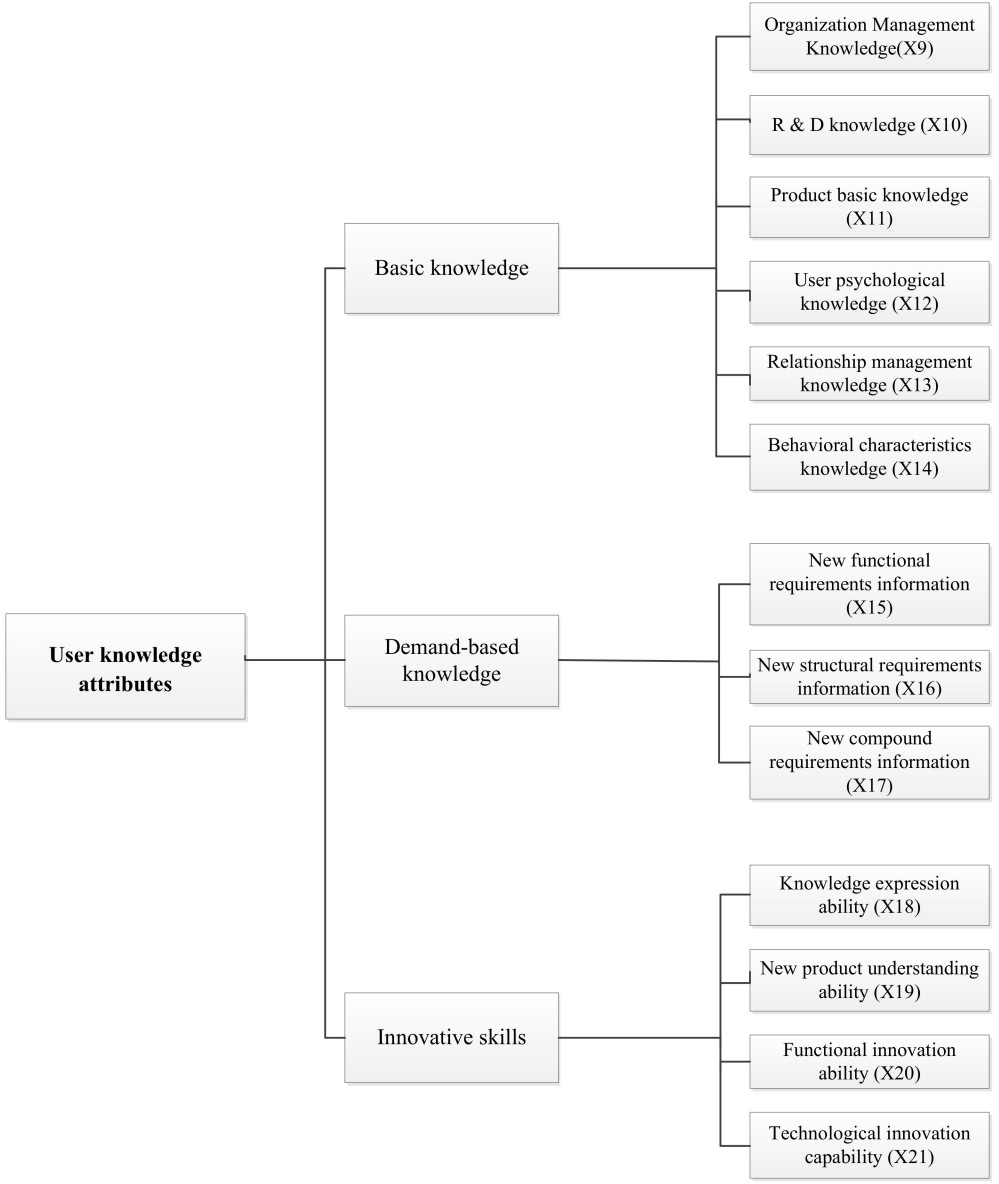

**Figure 1.** The factors of user knowledge attributes.

It is difficult to quantify the value of user knowledge attributes by using the traditional quantitative analysis method. This paper proposes a quantitative method for user knowledge attributes using a combination of linguistic variables and fuzzy mathematics [40]. By this method, this paper defines the linguistic variable set as $E = \{Very\ low,\ low,\ Medium,\ High,\ Very\ high\}$, and its corresponding value set as $S = \{0\sim2, 0.2\sim0.4, 0.4\sim0.6, 0.6\sim0.8, 0.8\sim1\}$. In this way, the fuzzy information can be digitized and the relevant input variables can be described quantitatively. Through expert interviews or test assessments, this paper can obtain data about user knowledge attributes. Since the values are all within the range of [0, 1], no further normalization is needed.

In summary, we can build a set of factors $X = \{x_1, x_2, \cdots, x_{20}, x_{21}\}$ for lead user identification, as shown in Table 1.

**Table 1.** Factor set for lead user identification.

| Factor Attributes | Factors | Variable Definition |
|---|---|---|
| User general attributes | Gender | $X_1$ |
| | Age | $X_2$ |
| | Monthly income | $X_3$ |
| | Number of owned products of a brand | $X_4$ |
| | Investment in purchasing products of a brand | $X_5$ |
| User activity attributes | Recency | $X_6$ |
| | Frequency | $X_7$ |
| | Monetary value | $X_8$ |
| User knowledge attributes | Organization management knowledge | $X_9$ |
| | R&D knowledge | $X_{10}$ |
| | Product basic knowledge | $X_{11}$ |
| | User psychological knowledge | $X_{12}$ |
| | Relationship management knowledge | $X_{13}$ |
| | Behavioral characteristics knowledge | $X_{14}$ |
| | New functional requirements information | $X_{15}$ |
| | New structural requirements information | $X_{16}$ |
| | New compound requirements information | $X_{17}$ |
| | Knowledge expression ability | $X_{18}$ |
| | New product understanding ability | $X_{19}$ |
| | Functional innovation ability | $X_{20}$ |
| | Technological innovation capability | $X_{21}$ |

## 3. The ICS-SVM Method for Lead User Identification

### 3.1. Support Vector Machine

The support vector machine (SVM) is a kind of learning algorithm based on the statistical learning theory proposed by Vapnik et al. [41]. This method is based on the principle of minimizing structural and empirical risk to improve the generalization ability of the algorithm. For the binary classification problem, the standard SVM assumes that the known sample set is $(x_i, y_i)$, $i = 1, 2, \ldots, l$, $x_i \in R^n$, $y_i \in \{-1, 1\}$, and the classification surface equation is:

$$w \bullet x + b = 0 \tag{4}$$

Therefore, the learning problem can be transformed into a minimized objective function:

$$\min \phi(w) = \tfrac{1}{2}\|w\|^2 + C\left(\sum_{i=1}^{l} \xi_i\right)$$
$$s.t. \quad y_i[(w \bullet x_i) + b] \geq 1 - \xi_i, \tag{5}$$
$$\xi_i \geq 0, i = 1, 2, \cdots, l.$$

where $C$ is the penalty factor, which is used to adjust the balance between structural risk and empirical risk, and $\xi_i$ is the empirical cost relaxation variable, which is used to allow misclassification in the case of linear inseparability. The above optimization problem can be solved by introducing the Lagrange function, whose solution process can be found in [42]. After solving the above problem, the optimal classification function can be obtained as:

$$f(x) = sign\left\{\sum_{i=1}^{l} \alpha_i y_i k(x \bullet x_i) + b\right\} \tag{6}$$

where $\alpha_i$ is the Lagrange multiplier corresponding to each sample and $k(x \bullet x_i)$ is the Kernel function, which is used to transform a linear inseparable problem in the low-dimensional input space into the high-dimensional feature space under the case of linear inseparability.

The traditional classification algorithm assumes that the importance of classification samples is the same, and the cost of misclassification is also the same [43]. The improvement

of classification efficiency is only achieved by increasing the prediction accuracy of the algorithm. With the expanding application of SVM and its continuous research, researchers have found that the cost of misclassification of different samples varies greatly in contexts such as disease diagnosis and credit evaluation [31,33], which makes it impossible to identify more meaningful samples during prediction. Therefore, the cost of misclassification is regarded as an important factor that cannot be ignored.

The cost-sensitive learning theory was put forward in this background and has been widely used in machine learning algorithms, which have become one of the most active and important research areas in machine learning [35]. The application of cost-sensitive learning theory can be divided into two categories: (1) the cost-sensitive meta-learning method, which transforms a non-cost-sensitive learning classification algorithm into a cost-sensitive learning algorithm, such as cost-sensitive BP neural networks; and (2) the direct method, which involves constructing a cost-sensitive classification algorithm, such as the cost-sensitive support vector, directly. The direct method has a stronger generalization ability than the cost-sensitive meta-learning method because it does not need an over-sampling or an under-sampling in the samples. Therefore, this paper uses the direct method.

### 3.2. Lead User Identification Model

The mathematical description of the lead user identification model based on ICS-SVM can be described as follows. $Y(i)$ is the output variable, which is used to indicate the status of classification result of user $i$; $X'(i) = (x1(i), x2(i), \ldots, xn(i))$ is the input variable, which is used to indicate the set of factors for classifying and identifying user $i$. The output variable represents the final goal, which is the result of users' classification in lead user identification. Therefore, the classification prediction output variable of user $i$ can be described as:

$$Y(i) = \begin{cases} 1 & \textit{Lead users} \\ -1 & \textit{Ordinay users} \end{cases} \tag{7}$$

$X'(i)$ is used as the input variable of the ICS-SVM, and $Y(i)$ is used as the output variable, which can form a sample couple $(X'(i), Y(i), cost(i))$, where $cost(i)$ is the cost of misclassification, which depends on the type of sample and the value of user. Then, the sample couples are trained and learned by the ICS-SVM for so that it can grasp the influence of all factors on lead user identification and the non-linear relationship among them. The model learns and outputs the predicted results which indicate whether the user meets the requirements of lead user.

Therefore, the learning problem can be transformed into solving the objective function as

$$\begin{aligned} \min \phi(w, \xi) &= \tfrac{1}{2}\|w\|^2 + C\left(\sum_{i=1}^{l} cost(i)\xi_i\right) \\ s.t. \quad & Y(i)[(w \bullet X'(i)) + b] \geq 1 - \xi_i, \\ & \xi_i \geq 0, i = 1, 2, \cdots, l. \end{aligned} \tag{8}$$

The cost of sample misclassification in user classification is integrated into the SVM objective function. The cost-sensitive support vector machine can be obtained by solving the optimization of (8). To solve the above optimization problem, the constrained optimization problem is transformed into the unconstrained optimization problem, and the Lagrange function can be established as:

$$L(w, b, \xi_i) = \frac{1}{2}\|w\|^2 + C\left(\sum_{i=1}^{l} cost(i)\xi_i\right) - \sum_{i=1}^{l} \alpha_i\{Y(i)(X(i) \bullet w + b) - 1 + \xi_i\} - \sum_{i=1}^{l} \mu_i\xi_i \tag{9}$$

where $\alpha_i \geq 0$ and $\mu_i \geq 0$ are Lagrange coefficients. To solve the minimization, the partial derivatives of $w$, $b$, and $\xi_i$ all need to be computed, and the partial derivatives should be 0.

$$\frac{\partial L}{\partial w} = w - \sum_{i=1}^{l} \alpha_i Y(i) X(i) = 0 \tag{10}$$

$$\frac{\partial L}{\partial b} = -\sum_{i=1}^{l} \alpha_i Y(i) = 0 \tag{11}$$

$$\frac{\partial L}{\partial \xi_i} = cost(i)C - \alpha_i - \mu_i = 0 \tag{12}$$

Substituting Equations (10)–(12) into Equation (9), the dual form of the optimization problem can be obtained as:

$$
\begin{aligned}
L_D &= \frac{1}{2}\sum_{i=1}^{l}\sum_{j=1}^{l} \alpha_i \alpha_j Y(i)Y(j) - \sum_{i=1}^{l} \alpha_i \\
s.t. \quad &\sum_{i,j=1}^{l} \alpha_i Y(i) = 0, \\
&0 \le \alpha_i \le cost(i)C, i = 1, \cdots, l
\end{aligned}
\tag{13}
$$

The optimal solution $\alpha^* = \left(\alpha_1^*, \cdots, \alpha_l^*\right)^T$ can be obtained by solving Equation (13), and the optimal weight $w^* = \sum_{i=1}^{l} \alpha_i^* Y(i) X(i)$ can be obtained by substituting $\alpha^* = \left(\alpha_1^*, \cdots, \alpha_l^*\right)^T$ into Equation (14). Choosing an optimal solution and sample that satisfy $0 < \alpha_i^* < cost(i)C$, the optimal bias $b^*$ can be obtained according to KKT condition. Therefore, after the combination kernel function is introduced, the optimal classification function in the nonlinear case can be obtained as:

$$f(x) = sign\left\{ \sum_{i=1}^{l} \alpha_i^* Y(i) K(X, X(i)) + b^* \right\} \tag{14}$$

where $K(X, X(i))$ is the combined weighted kernel function of the global kernel function and the local kernel function. The polynomial kernel function is a typical global kernel function, and the radial basis kernel function is a typical local kernel function. According to the linear combination formula of the kernel function in [41], we can obtain:

$$K(X, X(i)) = \lambda(X^T X(i) + 1)^q + (1 - \lambda)e^{-\|X - X(i)\|^2/(2\sigma^2)}, \lambda \ge 0 \tag{15}$$

where $\lambda$ is the combined kernel function weighting coefficient, $q$ is the polynomial kernel function parameter, and $\sigma$ is the radial basis kernel function parameter. All of these are particularly important. This paper optimizes the above parameters by a PSO algorithm [44], which can improve the recognition rate of minorities and balance the gap of samples. Finally, the cost of misclassification is reduced and the learning efficiency and prediction accuracy are improved. If the above parameters generated by a particle make the combined kernel function of the cost-sensitive support vector machine obtain a lower misclassification cost, the recognition rate of minority samples with higher cost of misclassification in lead user identification improves, and the value of the fitness function is reduced. By comprehensively considering the lead user misidentification rate and the cost of misidentification (the smaller, the better), we use the average value of the misidentification cost of the learning samples and the identification error rate under K-fold cross validation (K-CV) on the training set as the particle fitness function. Then, we can obtain the fitness function as following:

$$Fitness = \frac{\sum_{i=1}^{m} F(i)cost(i) \bullet \sum_{i=1}^{m} F(i)}{m^2} \tag{16}$$

where *m* is the number of samples in the test group and *F*(*i*) represents whether the users in the sample have been misclassified, which is represented by the Boolean variable. If it is 1, it means that user *i* is misclassified; otherwise, it is 0.

$$F(i) = \begin{cases} 1 & User\ identification\ is\ wrong \\ 0 & User\ identification\ is\ correct \end{cases} \tag{17}$$

In summary, the framework of the lead user identification model based on the ICS-SVM is shown in Figure 2.

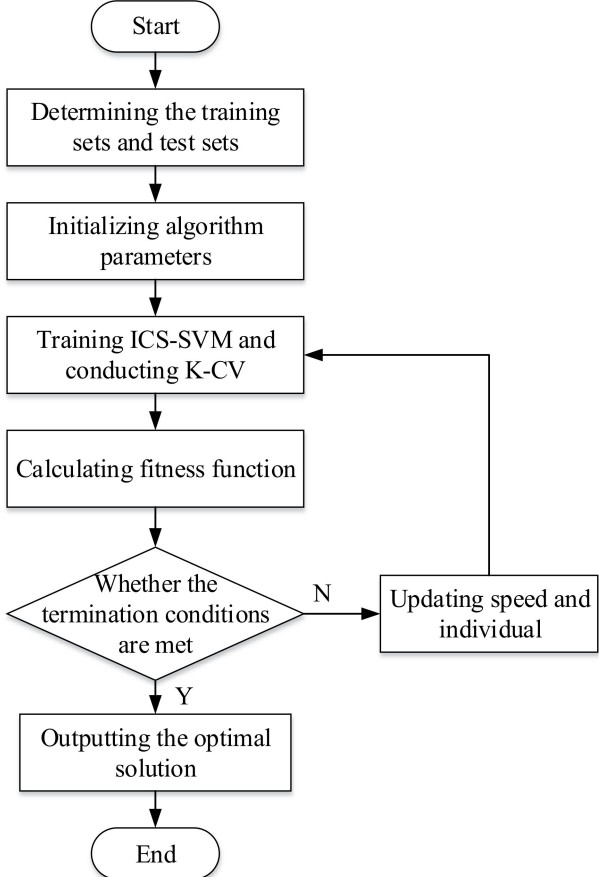

**Figure 2.** Framework of lead user identification model based on ICS-SVM.

### 4. Case Study

*4.1. Sample Data*

X is one of the most creative companies in China, focusing on the development of intelligent electronic products. It has achieved great success in adopting the CCPI mode. The proposed ICS-SVM method is used to identify the lead users among 96 users. Among the 96 users, there are 13 lead users and 83 ordinary users. The sample data contains the input data of lead user identification factors and the identification results as shown in Table 2.

**Table 2.** Sample data of all users.

|  | $X_1$ | $X_2$ | $X_3$ | $X_4$ | $X_5$ | $X_6$ | $X_{17}$ | $X_{18}$ | $X_{19}$ | $X_{20}$ | $X_{21}$ | $Y$ |
|---|---|---|---|---|---|---|---|---|---|---|---|---|
| 1 | 1 | 0.143 | 0.05 | 0.286 | 0.357 | 0.6 | 0.5 | 0.5 | 0.1 | 0.7 | 0.5 | −1 |
| 2 | 1 | 1 | 0.55 | 0.143 | 0.589 | 0.167 | 0.2 | 0.7 | 0.1 | 0.2 | 0.1 | −1 |
| 3 | 1 | 0.857 | 0.35 | 0 | 0 | 0.133 | 0.3 | 0.6 | 0.4 | 0 | 0.5 | −1 |
| 4 | 1 | 1 | 0.4 | 0.571 | 0.714 | 0.3 | 0.6 | 0.8 | 0.1 | 0 | 0.1 | −1 |
| 5 | 1 | 0.857 | 0.45 | 0.857 | 0.929 | 0 | 0.8 | 0.7 | 0.8 | 0.4 | 0.8 | 1 |
| 92 | 1 | 0.857 | 0.75 | 0.857 | 0.857 | 0.2 | 0.7 | 0.2 | 0.4 | 0.8 | 0.5 | 1 |
| 93 | 1 | 1 | 0.95 | 0.429 | 0.214 | 0.467 | 0.3 | 0.2 | 0.5 | 0.3 | 0.4 | −1 |
| 94 | 1 | 0 | 0.05 | 0.571 | 0.571 | 0.3 | 0.1 | 0.3 | 0.5 | 0.4 | 0.2 | −1 |
| 95 | 0 | 0.286 | 0.9 | 0 | 0.054 | 1 | 0.1 | 0.1 | 0.3 | 0.6 | 0.5 | −1 |
| 96 | 1 | 0.286 | 0.5 | 0.857 | 0.036 | 0.533 | 0.4 | 0.3 | 0.2 | 0.5 | 0.5 | −1 |

### 4.2. Lead User Identification Using ICS-SVM Method

From the sample data, it can be concluded that the ratio of ordinary users to lead users is about 6:1, which is an unbalanced sample. The sample data is randomly divided into two groups. One is the training group, including 80 sets of user data (8 lead users and 72 ordinary users), and the other is the test group.

To facilitate the statistics and calculations, the correct identification costs are set as 0. For the misidentification costs, the individual differences of users are ignored. Specifically, the misidentification cost that an ordinary user is misclassified as a lead user is named as $cost_1$ and the misidentification cost that a lead user is misclassified as an ordinary user is named as $cost_2$. According to the ratio of lead user and ordinary user samples and the two types of misidentification cost ratios suggested by experts, this paper takes $cost_1 = 1$ and $cost_2 = 6$.

The ICS-SVM algorithm is modified and improved on the basis of LibSVM 2.82 and then run by Matlab2010a. The PSO initial parameters are set as follows: particle number $M = 50$, learning factor $c1 = c2 = 1.8$, maximum number of iterations $kmax = 100$, particle dimension $D = 4$, preset minimum fitness value of PSO $Fitmin = 0$, initial particle position $S$, penalty factor $C = 3.5$, combined kernel function weighting coefficient $\lambda = 0.5$, polynomial kernel function parameter $q = 3$, and radial basis kernel function parameter $\sigma = 10$.

The ICS-SVM algorithm is run. When the number of iterations is $k = 183$, the lowest fitness value is achieved as $Fitness = 1.5625 \times 10^{-4}$. Meanwhile, for the optimal solution particle, the penalty factor $C = 3$, the combined kernel function weighting coefficient $\lambda = 0.46$, and the polynomial kernel function parameter $q = 2.835$, and the radial basis kernel function parameters $\sigma = 27.64$.

### 4.3. Comparison of Leading Customer Identification

The trained ICS-SVM is used to predict the test group and compared with the BP neural network and the RS-SVM method [45]. The BP neural network contains a hidden layer, and the RS-SVM kernel function and initial parameters adopt the same parameters as the ICS-SVM. This paper uses five indicators to compare the identification effect: the overall correct prediction rate $Z$, the weighted overall correct prediction rate $WZ$, the first type error rate $FZ_1$, the second type error rate $FZ_2$, and the average cost $EC$. Table 3 shows the confusion matrix of lead user identification to facilitate the statistics of the sample numbers of the four classification results and calculate the above five indices.

**Table 3.** Confusion matrix of identifying lead user.

| Classification Group Cost | Sample Group<br>Lead User | Ordinary User |
|---|---|---|
| Lead user | TN | FP |
| Ordinary user | FN | TP |

The overall correct prediction rate:

$$Z = \frac{TN + TP}{TN + FP + FN + TP} \tag{18}$$

The weighted overall correct prediction rate:

$$WZ = \frac{cost_1 TN + cost_2 TP}{cost_1(TN + FP) + cost_2(FN + TP)} \tag{19}$$

The first type error rate:

$$FZ_1 = \frac{FN}{FN + TP} \tag{20}$$

The second type error rate:

$$FZ_2 = \frac{FP}{TN + FP} \tag{21}$$

The average cost:

$$EC = \frac{cost_2 FN + cost_1 FP}{TN + FP + FN + TP} \tag{22}$$

The compared results of the different models are shown in Table 4.

**Table 4.** The compared results of different models.

|  | *Z* | *WZ* | *FZ$_1$* | *FZ$_2$* | *EC* |
|---|---|---|---|---|---|
| BP neural network | 87.5% | 82.93% | 20% | 9.09% | 43.75% |
| RS-SVM | 93.75% | 85.37% | 20% | 0 | 37.5% |
| ICS-SVM | 93.75% | 97.56% | 0 | 9.09% | 6.25% |

As can be seen from Table 4, in terms of *Z*, *WZ*, *FZ$_2$*, and *EC*, the ICS-SVM method proposed in this paper and the RS-SVM method have obvious advantages over BP neural networks. Compared with the RS-SVM, the ICS-SVM has the same overall prediction accuracy rate *Z*, the RS-SVM's *FZ$_2$* is better than that of the ICS-SVM, and other indicators of the ICS-SVM are higher than the RS-SVM. Firms that adopt the CCPI mode hope to improve the accuracy of lead user identification. Therefore, the comparison results of the above identification effects show that the ICS-SVM effectively avoids over-learning of large samples under sample imbalance conditions, and achieves better identification results, which verifies that the proposed the ICS-SVM method is effective to achieve successful lead user identification in CCPI.

## 5. Conclusions

Lead users are the most valuable kind of users, with pre-sighted product knowledge in new product development for firms. Therefore, the identification of lead users plays an important role for the success of CCPI. Based on the theory of cost-sensitive learning and support vector machines, this paper proposes a novel lead user identification method, which can improve the effectiveness and efficiency of lead user identification and user management in the CCPI mode. The main contributions of this work are as follows. First, the characteristics of lead user identification in CCPI are analyzed in-depth, and a factor set of lead user identification is proposed from multiple levels. Second, based on the characteristics of lead user identification, considering the sample misidentification cost and identification accuracy rate, an improved cost-sensitive learning support vector machine (ICS-SVM) model with PSO parameter optimization is established to achieve more convenient, effective, and reliable results in the process of leading user identification, which overcomes the shortcomings of traditional identification methods that are more

subjective and easy to misjudge. Third, taking a typical CCPI firm as an example to verify the effectiveness of the proposed method, the ICS-SVM method is shown to be able to effectively solve the different costs of unbalanced data and sample misidentification, and improves the identification performance from the perspective of misidentification cost and identification accuracy. With the help of the conclusions of this work, firms can quickly, efficiently, and cost-effectively identify lead users from massive user data in CCPI.

In future studies, we will conduct cross-industry comparisons and introduce this lead user identification method in virtual product and tangible product communities to improve the universality of this research. Secondly, considering that procedural and distributed fairness also play a significant role in lead user research, future research will investigate the ethical aspects of lead user identification in detail. Furthermore, we will try to apply the ICS-SVM lead user identification method in different user communities, especially in the big data context, to verify and improve its scalability and universality.

**Author Contributions:** Conceptualization, J.S., and X.C.; methodology, F.L.; software, F.Z.; formal analysis, N.Z.; investigation, F.Z.; resources, F.L.; writing—original draft preparation, J.S. and F.Z. All authors have read and agreed to the published version of the manuscript.

**Funding:** This research was funded by the Team of Trade Circulation Funding Project, grant number CJSYTD201701.

**Data Availability Statement:** Not applicable.

**Acknowledgments:** The authors are grateful to the support of the Doctoral Project of Chongqing Federation of Social Science Circles (2018BS84), Youth Foundation of Ministry of Education of China (19YJC630141), and the Chongqing humanities and social sciences research project (20SKGH110).

**Conflicts of Interest:** The authors declare no conflict of interest.

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
