# Peer review of "An Intelligent Method for Lead User Identification in Customer Collaborative Product Innovation"

_jtaer, doi:10.3390/jtaer16050088_

Round 1
Reviewer 1 Report
I agree with the responses, which are well documented. Also, the paper is very well revised, and I predict that it will have a good impact. The validation is especially appreciated. The corresponding changes and refinements have been made in the revised paper in accordance with the reviewer points.
Author Response
Response to Reviewer 1 Comments
Point: I agree with the responses, which are well documented. Also, the paper is very well revised, and I predict that it will have a good impact. The validation is especially appreciated. The corresponding changes and refinements have been made in the revised paper in accordance with the reviewer points.
Response: Thanks for your acceptance, and Thanks great for your valuable suggestions. These suggestions are really very important and helpful to improve our research.
Reviewer 2 Report
The adjective should come before the noun, in "For customer collaborative product innovation (CCPI), " Thus, it should be "collaborative customer product innovation (CCPI) [the acronym remains the same, however]
Noun-verb disagreement: "Therefore, if firms wants " I THOUGHT THIS REVISION WAS EXMAINED BY A NATIVE SPEAKER.
p. 3 "On the other hand" means that there was an "On the one hand..." introduced earlier for parallel construction. I do not see that. Re-word.
It is not my role to edit for grammar: "the opportunity cost that firm loses" ; this clause lacks a definite or indefinite article before the word "firm".
This makes no sense: "n cost inconsistency I this work[34, 35]. "
This not a logical sequence: Very low low Higher High Very high
There is a word(s) missing here: "If the above parameters generated by a particle make the combined kernel function cost-sensitive support vector machine obtain lower misclassification cost, (p. 11)
More definite/indefinite article issues: "play important role ..."
Author Response
Response to Reviewer 2 Comments
Point 1: The adjective should come before the noun, in "For customer collaborative product innovation (CCPI), " Thus, it should be "collaborative customer product innovation (CCPI) [the acronym remains the same, however]
Response 1: Thanks for the suggestion. We agree with that the adjective should come before the noun. However, customer collaborative product innovation (CCPI) is a generally accepted definition in the literature.
Point 2: TNoun-verb disagreement: "Therefore, if firms wants " I THOUGHT THIS REVISION WAS EXMAINED BY A NATIVE SPEAKER.
Response 2: Thanks for the careful suggestion, and really sorry for the careless. We have revised this error, and we have also carefully checked and revised the whole manuscript to avoid the similar errors.
Point 3: p. 3 "On the other hand" means that there was an "On the one hand..." introduced earlier for parallel construction. I do not see that. Re-word.
Response 3: Thanks for the careful suggestion, and really sorry for the careless. We have revised this issue.
Point 4: It is not my role to edit for grammar: "the opportunity cost that firm loses" ; this clause lacks a definite or indefinite article before the word "firm".
Response 4: Thanks for the careful suggestions and kindness. We have revised this issue, and we have also carefully checked and revised the whole manuscript to avoid the similar errors.
Point 5: This makes no sense: "n cost inconsistency I this work[34, 35]. "
Response 5: Thanks for the careful suggestion. It should be “in this work[34, 35].”, and we have revised it.
Point 6: This not a logical sequence: Very low low Higher High Very high
Response 6: Thanks for the careful suggestion. Yes, it is not a logical sequence, and we have revised it as “Very low, low, Medium, High, Very high”.
Point 7: There is a word(s) missing here: "If the above parameters generated by a particle make the combined kernel function cost-sensitive support vector machine obtain lower misclassification cost, (p. 11)
Response 7: Thanks for the careful suggestion. We have revised this sentence as “If the above parameters generated by a particle make the combined kernel function of cost-sensitive support vector machine obtain lower misclassification cost”
Point 8: More definite/indefinite article issues: "play important role ..."
Response 8: Thanks for the careful suggestions and kindness. We have revised it as “play an important role ...”, and we have also carefully checked and revised the whole manuscript to avoid the similar errors.
This manuscript is a resubmission of an earlier submission. The following is a list of the peer review reports and author responses from that submission.
Round 1
Reviewer 1 Report
Some of the English syntax is awkward. Notice the redunancy: "Identification of lead users is to identify ....'
Avoid etc. ALWAYS. And then, we need a new paragraph here: "etc. Scholars...."
NEVER use two i.e., in the same sentence or in close proximity. "the lead user characteristics[23], i.e. who to search for, and the identification methods[1, 24], i.e. how to identify lead users. "
A new paragraph is needed with "Among" "groups can be identified more quickly[28, 30]. Among the intelligent machine learning"
Missing the word "variable" here: "The Recency indicates...." Continue with the inclusion of the word "variable" after all the key variables that follow (By the way: There should be line numbers here so I can refer the readers to the specific section. This is NOT the author's fault).
Figure 1 misspells Behavioral in one of the boxes. This paper should have been edited more carefully. The EDITOR/PUBLISHING house should ensur this and then kick it back to the author. MDPI?
[MISSING: "The..."The rraditional classification algorithm assumes..."
The conclusions are speculative and are not commensurate with the effort it takes to read both the turgid English of the paper as well as the conclusions, which follow: "Based on the theory of cost-sensitive learning and support vector machine, this paper proposes a novel ICS-SVM lead user recognition identification method, which can improve the identification efficiency of lead users, and improve the efficiency of user management in CCPI mode. The main contribution of work is as follows: First, the characteristics of lead user identification in CCPI are analyzed in-depth, and a factors set of lead user identification is proposed from multiple levels. Second, based on the characteristics of lead user identification, considering the sample misidentification cost JTAER 2021,16,FOR PEER REVIEW 16 and identification accuracy rate, an improved cost-sensitive learning support vector machine (ICS-SVM) model with PSO parameter optimization was established, which overcomes the shortcomings of traditional identification methods that are more subjective and easy to misjudge. Third, taking a typical CCPI firm as an example to verify the effectiveness of the proposed method, the ICS-SVM method can effectively solve the different costs of unbalanced data and sample misidentification, and improves the identification performance from the perspective of misidentification cost and identification accuracy."
The reader deserves much more.
Reviewer 2 Report
This study has not developed a research gap in the introduction and fails to justify; however, even after changes made to the manuscript. I believe that paper could be improved. I suggest major revision.
Your paper title serves as the initial guide to the essence of your work, so please revise your title, so it includes the most important elements of your report.
Exploring the effect of buyer engagement on green product innovation: Empirical evidence from manufacturers: DOI: https://doi.org/10.1002/bse.2631
Role of Design Thinking and Biomimicry in Leveraging Sustainable Innovation
https://doi.org/10.1007/978-3-319-71059-4_86-1
- The abstract is too long. The idea is to clearly and briefly provide the research gap, methodology, main objective, key contributions, and major results.- Please revise the conclusion in the abstract to avoid overly casual language. Revise your abstract to include methods employed and implications.
-Introduction part still needs major changes. It seems that the authors have highlighted the Different definitions. I personally think it does not have relevance to explain different definitions here. Moreover, I suggest authors review the importance of core idea and how does it has been researched previously. Also, explain, what previous literature review studies or metaanalysis studies have been carried out, and what was missing in the previous studies? How this study will fill this gap.
Effects of buyer-supplier relationship on social performance improvement and innovation performance improvement https://doi.org/10.1504/IJAMS.2019.096657
- There is no flow in the text. It partly depends on the lack of proofreading but also on the fact that many statements and claims are made without being followed up by a clear and logical discussion. It is especially problematic in the introduction that brings up several findings from different areas without linking them together
The Role of Technological Innovation in a Dynamic Model of the Environmental Supply Chain Curve: Evidence from a Panel of 102 Countries
https://doi.org/10.3390/pr8091033
-Also explain briefly what the future research opportunities are
-The authors should demonstrate how the results of the study could be transferred to other
contexts, and to what extent the manuscript contributes to the literature